# Effects of Laboratory Ageing on the FTIR Measurements of Water-Foamed Bio-Fluxed Asphalt Binders

**DOI:** 10.3390/ma16020513

**Published:** 2023-01-05

**Authors:** Marek Iwański, Anna Chomicz-Kowalska, Krzysztof Maciejewski, Mateusz M. Iwański, Piotr Radziszewski, Adam Liphardt, Jan B. Król, Michał Sarnowski, Karol J. Kowalski, Piotr Pokorski

**Affiliations:** 1Department of Transportation Engineering, Faculty of Civil Engineering and Architecture, Kielce University of Technology, Al. Tysiąclecia Państwa Polskiego 7, 25-314 Kielce, Poland; 2Department of Building Engineering Technologies and Organization, Faculty of Civil Engineering and Architecture, Kielce University of Technology, Al. Tysiąclecia Państwa Polskiego 7, 25-314 Kielce, Poland; 3Institute of Road and Bridges, Faculty of Civil Engineering, Warsaw University of Technology, Al. Armii Ludowej 16, 00-637 Warsaw, Poland

**Keywords:** foamed bitumen, fluxing agent, polymer modified asphalt binder, highly modified asphalt binder, short-term ageing, long-term ageing, FTIR

## Abstract

The study investigated the effects of laboratory ageing on the fluxed and water-foamed asphalt binders in scope of Fourier transform infrared spectroscopic measurements of ageing indicators and changes in their chemical composition. The investigated binders included two paving grades, two polymer modified asphalt binders, and a highly modified asphalt binder. The bio-flux additive was produced from rapeseed methyl esters in an oxidation reaction in the presence of a metal catalyst and organic peroxide. The use of the bio-origin additive, in particular oil derivatives, was aimed at softening and better foaming of asphalt binders. This modification is possible due to the good mixability of vegetable oils with an asphalt binder, which gives a homogeneous product with reduced stiffness. The study involved the rolling thin film oven, short-term, and the pressure ageing vessel, long term, and ageing to induce oxidation on the evaluated asphalt binders. The addition of the bio-flux additive has significantly decreased the measured content of ketone compounds related to oxidation in both non-aged and aged asphalt binders, although this effect after ageing were far smaller in magnitude. Additionally, both ageing processes decreased significantly the absorbances in the ester spectral bands specific to the bio-flux additive. All mentioned effects were similar in magnitude in all tested asphalt binders.

## 1. Introduction

The main source of emissions and energy intensity of the production of asphalt mixtures include heating the mixture constituents and allowing their adequate mixing and compaction in the field [1,2]. Decreasing the processing temperatures of asphalt mixtures by introducing warm and half-warm mix asphalt techniques may significantly reduce the effects of pavement construction on the environment. By doing so, emissions of greenhouse gasses, organic volatile compounds along with the energy consumption of the process can be significantly reduced [3,4].

The commonly used methods for producing warm-mix asphalt include the utilization of proprietary techniques and additives (e.g., liquid and solid warm mix additives) [5,6,7], asphalt binder foaming using zeolite and other water-bearing minerals [8,9,10,11,12], direct water injection [13,14,15,16] and use of bio-based fluxing agents [17,18,19,20,21]. In the recent years techniques enabling production of half-warm mix asphalt mixtures have also gained significant attention [22,23,24]. One of the added benefits of warm mix asphalt techniques is the possibility of extending the hauling distance of the mixture [25]. 

Warm mix asphalt techniques based on water-injection foaming base on the vaporization of water on contact with hot bitumen and subsequent formation of water vapor-bitumen foam. The decrease in processing temperatures of an asphalt mixture is possible due to the increased volume of the binder enhancing the mixture coatability and the shear-thinning mechanism promoted in the asphalt binder phase with a highly reduced binder film thickness in the foam [26]. A major advantage for implementing water-injection based foaming processes is in warm mix asphalt production is that once the asphalt plant is adequately modified (or a plant permitting asphalt foaming installed initially), no additional costs are required to produce asphalt mixtures at lowered temperatures compared to typical hot mix asphalt. Therefore, this method can be easily and affordably combined with other techniques to enhance the characteristics of the asphalt binder [9,27,28,29,30,31] and improve the performance of an asphalt mixture. 

In the recent years, the use of a number of different bio-based additives has been investigated to modify the properties of asphalt binders and asphalt mixtures, in some cases to facilitate their paving in lowered temperatures. Oleoflux, an additive based on fatty acid methyl esters from sunflower oil, was shown to lower the bituminous binder’s viscosity, promote the binder’s adhesion to the aggregate, and improve the moisture resistance of WMA asphalt mixtures [32]. Green Seal is an another plant-derived fluxing additive, consisting of a liquefied vegetable resin with monoalkyl esters from vegetable oils and animal fats and primarily used in low concentrations to improve binder adhesion [32]. At higher concentrations however, it softens the asphalt binder significantly, making it adequate in high RAP applications [32]. Used cooking (post-frying) oils, referred often as waste cooking oils (WCO), can also be used as liquefying additives and as rejuvenators [33]. Among bio-flux additive of plant origin distilled tall oils are also recognized, which are a by-product of paper production. Tall oils are used as emulsifiers, anti-stripping additives, additives to warm mix asphalt and as rejuvenators [21].

In the recent years, a range of new types of bio-flux additive has been introduced, which enables production of highly performing asphalt mixtures at lowered temperatures [34], mixtures with larger amounts of RAP without deteriorating their properties [35] that retain high fatigue life and resistance to atmospheric factors, such as low temperatures and water [36,37]. One such additive (Bioflux) [38] is produced from rapeseed methyl esters (RME), subjected to oxidation in the presence of a metal catalyst (acid salts of cobalt) and organic peroxide polymerization promoter (cumene hydroperoxide). The concept of the production and mechanism of the oxypolymerization reaction has been described in detail in [39,40]. Bioflux, when added to bitumen, lowers its viscosity, which allows for lowering technological temperatures and producing the asphalt mixture in the WMA (warm mix asphalt) technology initially, but with time it permits an increase in the performance of the binder and mixture [38,40]. It was found that the Bioflux additive has high potential to decrease the production temperatures of asphalt mixtures with a highly (elastomer) modified asphalt binder without sacrificing its exceptional high temperature performance [34]. In a different work, it was shown that highly modified asphalt binders are well suited for being foamed using the water-injection method [41]. A recent study [42] has investigated the effects of simultaneous use of fluxing using the Bioflux additive and water foaming different asphalt binders, including polymer modified and highly modified binders. It was shown that the response of the binders to fluxing differed greatly depending on the type of the binder, and the fluxed, highly modified asphalt binder improved its high-temperature performance after foaming. 

Fourier transform infrared spectroscopy (FTIR) is a popular technique to investigate changes in the chemical composition of asphalt binders subjected to thermal oxidation [43,44,45], ultraviolet ageing [46] water exposure [47] and other types of influence (e.g., subject to radioactive wastes [48]). Recently, this technique has been employed to identify temperature phase transitions in asphalt binders [49,50]. Particularly, this technique is effective in tracking the formation of carbonyl and sulfoxide oxidation products, which can be linked to both different stages of pavement service [51] and specific changes in the rheological characteristics of the binders [52]. Measurements on asphalt binders using FTIR method are typically performed in the attenuated total reflectance (ATR) mode due to the nature of the material and the mid-infrared is used [53]. The FTIR method is also used in characterization of other petroleum-based products, particularly biofuels due to the distinct responses produced by the bio-additives (fatty acid methyl esters) used in their composition [54,55].

Based on the state of the art and the findings of the recent studies in the field, a study has been conducted to investigate the effects of laboratory ageing on the bio-fluxed and water-foamed asphalt binders in the scope of Fourier transform infrared spectroscopic measurements of ageing indicators and changes in their chemical composition. The investigated binders included paving-grade and polymer-modified asphalt binders commonly used in paving in Poland. The study involved the rolling thin film oven test (RTFOT), short-term, and the pressure ageing vessel (PAV), long term, ageing to induce oxidative stress on the evaluated asphalt binders. 

## 2. Materials and Methods

### 2.1. Materials

#### 2.1.1. Asphalt Binders

Five petroleum asphalt binders were used in this study: 20/30 and 50/70 paving grade bitumens, 25/55–60 and 45/80–55 polymer modified asphalts and a 45/80–80 highly modified asphalt binder (HiMA). The asphalt binders were selected due to their wide use in asphalt paving in Poland [56], their differences in properties and degree of polymer modification. The asphalt binders were commercially sourced from Orlen Asfalt (Płock, Poland). The basic characterization of these base asphalt binders is presented in Table 1.

#### 2.1.2. Bio-Derived Bio-Flux Additive

The bio-flux additive was produced using pure fatty acid methyl esters (FAME) derived from rapeseed oil (rapeseed methyl esters, RME). The RME used in this study differed from the typical formulation by the absence of anti-ageing additives, as it was to be subjected to an oxidation reaction to alter its properties. On the contrary, fatty acid methyl esters used as an additive to biofuels contain anti-ageing additives, which, however, may block the hardening and stiffness recovery in asphalt binder and mixture. Some selected RME properties used in the study are shown in Table 2.

The pure RME’s were subjected to an oxidation reaction in the presence of oxidation promoters:− cobalt catalyst: 0.1 % m/m in converted to metal− polymerization initializer: cumene hydrogen peroxide 1.0% m/m

The reaction was conducted in a laboratory reactor with a 0.3 ratio of reactor diameter to RME height and with an initial temperature of 25 °C. The RMEs were oxidized for two hours with an airflow value of 500 L/h per 1 kg of the product. The final product was kept in a sealed steel container at 5 °C until before use.

### 2.2. Methods

#### 2.2.1. Design of Experiment

The experimental plan was set up to investigate the combined effects of laboratory ageing on the FTIR spectral responses of bio-fluxed and foamed asphalt binders. The FTIR measurements were done in triplicates for the five selected asphalt binders before ageing, after RTFOT ageing and after RTFOT+PAV ageing. The effects of the bio-flux additive were investigated in all mentioned cases based on the measurements of non-fluxed binders and using its three dosing rates: 1%, 2% and 3%, which were defined in preliminary testing (based on experiences from previous studies [35,39,40,42]).

In relevant cases, the asphalt binders (after adding bio-flux additive and asphalt foaming when applicable), asphalt binders were subjected to the rolling thin film oven test (RTFOT, EN 12607-1) and the pressure ageing vessel (PAV, EN 14769) ageing. 

#### 2.2.2. FTIR Measurements

The effects of laboratory ageing and bio-flux additive under investigation were evaluated by Fourier-transform infrared spectroscopy (FTIR) using the attenuated total reflectance (ATR-FTIR) method. The Thermo-Scientific Nicolet iS 5 FTIR Spectrometer (Waltham, MA, USA) and the PIKE Technologies GladiATR (Madison, WI, USA) attenuated total reflectance accessory with a diamond window was used. The changes in the chemical structure of the asphalt binders were tracked using structural chemical indices characterizing the relative changes in the amounts of compounds related to oxidative ageing (sulfoxide and carbonyl indices) and an index specific to the bio-flux modification. Details regarding the analysis of the spectra are provided in Section 3.2. The spectrograms were recorded using 32 scans per sample at 4 cm^−1^ resolution and three replicates were tested for each evaluated experiment.

Prior to any evaluations and calculations, all absorbance spectra have been normalized by setting the absorbance value of the asymmetric stretching vibration of the aliphatic structures at 2923 cm^−1^ to 1.0 and multiplying the entire spectrum by an adequate factor as proposed by Hofko et al. [53]. 

The statistical analysis for assessing the significance of the measured effects was conducted based on a linear statistical model with an interaction term. 

## 3. Results

### 3.1. Evaluation of the FTIR Spectra of the Asphalt Binders and Bio-Flux Additive

The FTIR spectra of the evaluated neat asphalt binders are presented in Figure 1. 

The typical areas of interest in the analysis of asphalt binders’ infrared spectrums include the absorption bands related to the formation of oxidation products and their possible polymer modification. 

The presence of spectral peaks in the carbonyl bands and sulfoxide bands inform us about the formation of different ageing-related compounds in bituminous binders. The absorbance responses in the 1700 cm^−1^ wavenumber region are related to the presence of anhydrides (1775–1725 cm^−1)^, esters (1750–1735 cm^−1^), carboxylic acids (1714–1700 cm^−1^), ketones (1700–1660 cm^−1^) and amides (1660–1640 cm^−1^), while the responses around the 1300 cm^−1^ wavenumber are characteristic to sulfoxides [57,58].

The absorbance spectra of the investigated asphalt binders differed the most in the absorption bands related to butadiene, styrene and vinyl structural groups. All polymer modified asphalt binders recorded pronounced absorption bands encompassing 966 cm^−1^ and 699 cm^−1^ wavenumbers, which can be related to the presence of polybutadiene and polystyrene, indicating a presence of a styrene-butadiene-styrene (SBS) copolymer [59]. In addition to this, the 45/80–80 asphalt binder exhibited significant absorption peaks at 910 cm^−1^ and 990 cm^−1^ wavenumbers, which could be related to the utilization of high vinyl SBS, which is reported to be used in HiMA formulations [60,61].

Figure 2 presents the FTIR spectrum obtained on a sample of the bio-flux additive.

The obtained spectra of the bio-flux additive exhibited typical characteristics of a rapeseed oil [55], with additional absorption spectrum bands at 1436 cm^−1^ and 1195 cm^−1^ wavenumbers. These responses are related the O-CH_3_ stretching and CH_3_- asymmetric bending, respectively, which are characteristic to fatty acid methyl esters and not present in their precursor oils [54]. A distinctive absorption band, typically found in bio-oil derivatives, was found at 1741 cm^−1^ wavenumber, corresponding to the stretching of the -C=O in the ester functional group. This band is used in quantification of FAME in different formulations, e.g., in petroleum based biofuels [54]. Figure 3 and Figure 4 present the carbonyl (a) and sulfoxide (b) regions of the overlayed spectra of asphalt binders and the bio-flux additive, selected to showcase the observed effects. 

The addition of the bio-flux additive to the asphalt binders resulted in the appearance of the 1741 cm^−1^ ester absorption peak, in which height increased with the increase of the additive’s concentration. Despite the fact that this absorption band is relatively broad and very high in intensity in pure bio-flux, its main effects observed response in the asphalt binders were restricted to approx. 1755–1730 cm^−1^ wavenumber range, recognized as the ester and anhydride absorption bands. Nevertheless, the presence of such strong absorbance bands affected some of the binder spectrums by increasing the absorbance intensities of the spectral lines in the proximity of the 1754–1730 cm^−1^ range. These effects can be seen as far as at 1770 cm^−1^ and 1700 cm^−1^ wave numbers, affecting the ketone specific absorption bands of the asphalt binders. 

Inspection of the FTIR spectra of asphalt binders subjected to laboratory ageing showed some effects of the bio-flux additive. The changes in the carbonyl area of the spectrums were generally similar as in other studies, showing that in the course of oxidative ageing of asphalt binders, esters are typically not produced, while anhydrides (1775–1725 cm^−1)^ and carboxylic acids (1714–1700 cm^−1^) are formed in only small amounts [51]. Ketones (1700–1660 cm^−1^), which concentrations may be an order of magnitude higher than the anhydrides [51], made up the majority of oxidation products which form absorption bands in the carbonyl region. As shown in Figure 4a, the major increases in the carbonyl region absorption bands due to laboratory ageing were restricted to the 1714–1660 cm^−1^ wavenumber region. Nevertheless, the RTFOT and PAV ageing caused changes in the bio-flux specific absorption band (1754–1730 cm^−1^), resulting in a decrease of its height and area. The observed absorption bands decreased initially after RTFOT and continued to decrease after the PAV procedure. 

The changes in the sulfoxide region (1030 cm^−1^) were primarily shaped by the employed ageing procedures, while the effects due to the addition of bio-flux were difficult to evaluate by studying and comparisons of the absorbance spectra.

### 3.2. Quantification of the FTIR Spectra Responses

#### 3.2.1. Evaluation of Methods for the Quantification of the FTIR Spectra Responses

The quantitative analysis of the FTIR spectra involved evaluation of chemical indices characterizing abundance of carbonyl (ketone), sulfoxide and ester structural and functional groups in the asphalt binders based on the areas of their characteristic absorption bands. 

Recent studies, or repeatability and reproducibility of FTIR measurements, have shown [45,53] that this method is not typically recommended for evaluation of ageing responses of asphalt binders, and an absolute baseline method should be adopted instead. However, in cases when evaluated absorption bands interfere and may be biased by the proximity of other absorption bands, the tangent baseline method provides more reliable results [53]. As described in Section 3.1 and shown in Figure 3a and Figure 4a, it is the case when the ester and ketone carbonyl bands are evaluated in asphalt binders modified with RMEs. Therefore, the areas under the absorption bands were computed by integration using the common tangent baseline method as described in [62]. 

One of the methods for calculating chemical indices for quantification of FTIR responses involves dividing the areas of absorption peaks in question, by reference values obtained from the same spectrum. These reference values should be obtained from an absorption band of a chemical group or multiple groups characterized by the least variation possible. Some of commonly used approaches include use of the aliphatic absorption bands (1525–1350 cm^−1^), which are typically not affected by ageing processes [63] or a sum of peak areas in wider ranges of the spectrum [52,58]. The results of the analysis of selected reference values are presented in Table 3.

Three methods for calculating reference values were evaluated: one based on the aliphatic group and two based on the sums of peak areas in the 2000–600 cm^−1^ and 2953–699 cm^−1^ wavenumber ranges. The calculations were done separately for each asphalt binder due to the differences in their absorption spectra. Table 3 shows that the smallest coefficients of variance of the evaluated areas were recorded in the approach (3) in case of all asphalt binders. This was due to the large sum of the integrated peaks, and relatively low variability measured by both the span of measured values (“max−min”) and standard deviations. Based on these results, in this study the approach utilizing the 2953–699 cm^−1^ wavenumber range was selected to obtain the unbiased values of the calculated chemical indices. 

The quantitative analysis of the FTIR absorption spectra involved evaluation of the presence of products of oxidation of mentioned compounds was conducted by calculating normalized indices as given in Table 4 based on the areas under the respective peaks.

#### 3.2.2. Effects on the FTIR Oxidative Ageing Indicators in Asphalt Binder

The effects of the bio-flux additive on the formation of oxidative ageing products were carried out by evaluating chemical indices corresponding to the prevalence of carbonyl (ketone) and sulfoxide structures in the asphalt binders subjected to short-term and long-term ageing. Figure 5 presents the calculated carbonyl indices in the evaluated asphalt binders and corresponding statistical analysis of the measured effects is shown in Table 5.

Quantification of the FTIR spectra has shown that the addition of the bio-flux additive had major effects on the measured absorbance bands in the carbonyl region corresponding to the formation of ketones. In most cases of the non-aged and RTFOT-aged binders, decreases of the carbonyl index proportional to the bio-flux additive content were recorded. In the case of the 25/66–60 polymer-modified, non-aged binder and the 20/30 RTFOT-aged binder, the magnitudes of these changes were far smaller in magnitude than in the remaining binders. Distinct observation was made in the case of the 45/80–80 highly modified asphalt binder in which the addition of bio-flux initially decreased significantly the values of I_C=O_ index but increasing the content of the additive to 2 and 3% resulted in only small changes in this respect. After long-term PAV ageing, the effects of the additive were less pronounced, and visible mostly at the higher end of its dosing rate (2 and 3%), where it decreased the measured amounts of carbonyl compounds by a small amount. Again, these effects after PAV ageing were the strongest in case of the 45/80–80 asphalt binder. 

It must be noted that oxidation is the main factor responsible for the aging of asphalt binders. Molecular oxygen can react with the binder’s reactive components, leading to the formation of oxygen compounds, among which ketones, carboxylic acids and sulfoxides predominate. The most intense oxidation of asphalt binder occurs during the mixing, production and laying of the asphalt mixture. Temperature and air access have the greatest impact on the rate of oxidation. For example, a temperature increase of 10 °C results in a twofold increase in the oxidation rate, so during operations with asphalt at elevated temperatures, aging occurs to a greater extent. Adding bio-flux additive to the asphalt binder reduces the viscosity of the asphalt binder, and the required process temperature can be reduced, which reduces ageing. In addition, bio-flux additive contains unsaturated bonds and is susceptible to the oxypolymerization reaction. This results in the fact that molecular oxygen is primarily involved in the oxypolymerization reaction of the reactive bio-additive, which limits the ageing of the asphalt binder.

The statistical analysis has shown that the bio-flux additive content and the ageing-state of the investigated asphalt binders had statistically significant (*p* < 0.05) effects on the calculated values of the carbonyl index. This included the significance of the interaction terms encompassing the joint effects of the additive and laboratory ageing. In general, the effects of bio-flux additive and the interaction effects between ageing and the additive on decreasing the I_C=O_ values were greatest in the 20/30 asphalt binder and smallest in the 25/55–60 binder. 

Figure 6 presents the calculated sulfoxide indices in the evaluated asphalt binders and corresponding statistical analysis of the measured effects is shown in Table 6.

The sulfoxide region of the FTIR spectra of the investigated asphalt binders was mostly unchanged by the addition of the bio-flux additive. Most of the variance seen in the data is comparable in magnitude to the 95% confidence intervals of the I_S=O_’s mean values. The effects of the additive in this respect were found to be statistically significant only in the case of the 25/55–60 and the 45/80–80 asphalt binders, but the significance of the interaction terms were not proven.

#### 3.2.3. Effects on the Measured Bio-Flux Additive Content

Figure 7 presents the calculated bio-flux related indices in the evaluated asphalt binders and corresponding statistical analysis of the measured effects is shown in Table 7.

The areas under the 1755–1730 cm^−1^ absorbance bands of the investigated binders were strongly correlated with the bio-flux additive content. Increase in dosing rate of the bio-flux additive resulted in approximately proportional growth of the peak areas related to the presence of ester functional groups. These relationships persisted in RTFOT and PAV aged asphalt binders, although the 1755–1730 cm^−1^ absorbance bands were significantly affected by the short- and long-term ageing. The ageing processes resulted in prominent decreases in the bio-flux related absorbance peaks of the FTIR spectra of all investigated asphalt binders. The greatest changes in this respect were observed in the case of the 25/55–60 asphalt binder after PAV ageing, while the smallest were seen in the 50/70 binder. The I_Bio_ values measured in the asphalt binders without the addition of bio-flux additive were close zero and were virtually not affected by the ageing protocols. 

The statistical analysis provided in Table 6 shows that the significant factors affecting the I_Bio_ responses in the investigated binders include the bio-flux additive content alone and its interaction with the ageing state of the binders. The fact that the main effects relating to the RTFOT and PAV registered high *p*-values (*p* > 0.05) were calculated for ageing, indirectly confirms that the absorbance band in the ester specific 1755–1730 cm^−1^ wavenumber region did not respond to the laboratory ageing.

Figure 8 presents the relationships between the added bio-flux additive and the I_Bio_ indices calculated based on the responses in the 1755–1730 cm^−1^ wavenumber absorption bands in the evaluated asphalt binders. The relationships between the amount of added bio-flux additive and measured I_Bio_ responses the in all non-aged binder were similar, with the exception of the 20/30 asphalt binder, which was characterized by an increased slope. The low variability of the intercepts and the slopes of the obtained regression lines shows that the bio-flux content in these binders could be determined with reasonable accuracy based on the FTIR measurements. In the case of the aged asphalt binders, the regression lines are mostly colinear, however a uniform spread in their slopes was observed. These effects could lead to errors in the estimation of bio-flux content amounting up to 1%. 

## 4. Conclusions

The conducted study investigated the FTIR responses of selected foamed asphalt binders containing a bio-flux additive based on oxidized rapeseed methyl esters in the scope of laboratory short- and long-term ageing conducted using the RTFOT and PAV procedures. The analyses included evaluation of the effects on thermal and oxidative stress on the FTIR indicators regarding the formation of carbonyl (ketone) and sulfoxide compounds in bituminous binders as well as the changes of ester-specific responses corresponding to the bio-flux additive. Based on the study, the following was concluded:the ester functional group absorption peak (1741 cm^−1^) present in fatty acid methyl esters was found to be suitable for identifying the presence of bio-oil derived bio-flux additive;the absorption peaks at the 1741 cm^−1^ wavenumber characteristic to the RME based bio-flux additive were reasonably confined to the 1755–1730 cm^−1^ absorption band and therefore separated from the absorption bands affected by laboratory oxidative ageing (RTFOT, PAV);at higher concentrations of the bio-flux additive, the broad and intense ester absorption band impacted the carbonyl region of the asphalt binders, and a tangent baseline integration method of calculating peak areas was used;the presence of bio-flux additive decreased the amount of identified carbonyl compounds (ketone) in the asphalt binders before and after ageing;sulfoxide content and formation during ageing was mostly unaffected by the presence of bio-flux additive;the measured prevalence of ester functional groups in asphalt binders containing bio-flux additive, established based on the FTIR measurements, decreased in the course of RTFOT and PAV ageing.

Based on the aforementioned findings, it can be stated that the presence of bio-flux additive has significantly affected the measured content of compounds related to oxidative stress in non-aged asphalt binders. The additive has also decreased their contents in laboratory-aged binders, although this effect was significantly less pronounced than in the case of the non-aged binders. These effects could contribute to the effects of fluxing action of the bio-additive and subsequent recovery of the binders’ stiffness as observed in [38,40,42].

## Figures and Tables

**Figure 1 materials-16-00513-f001:**
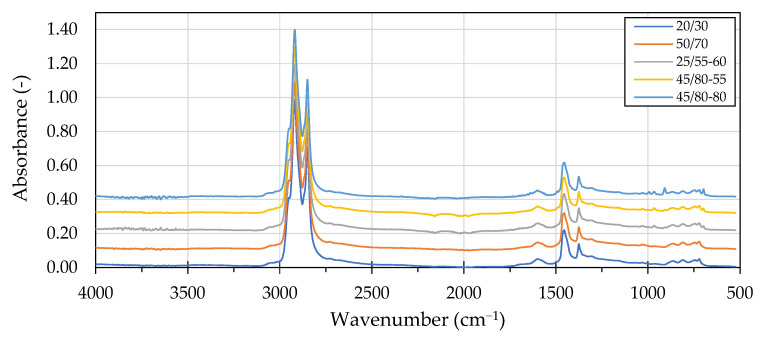
FTIR spectra of the investigated asphalt binders.

**Figure 2 materials-16-00513-f002:**
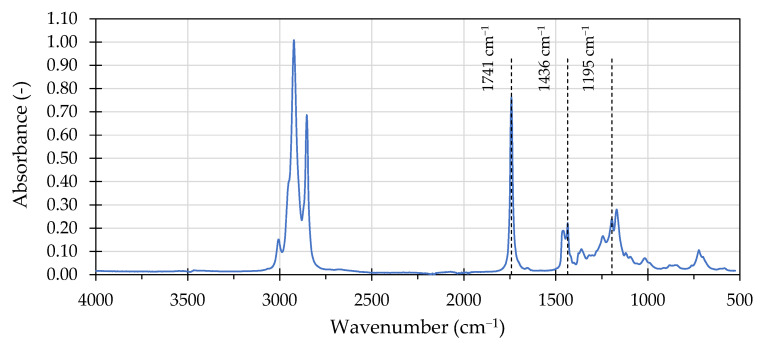
FTIR spectrum of the bio-flux additive.

**Figure 3 materials-16-00513-f003:**
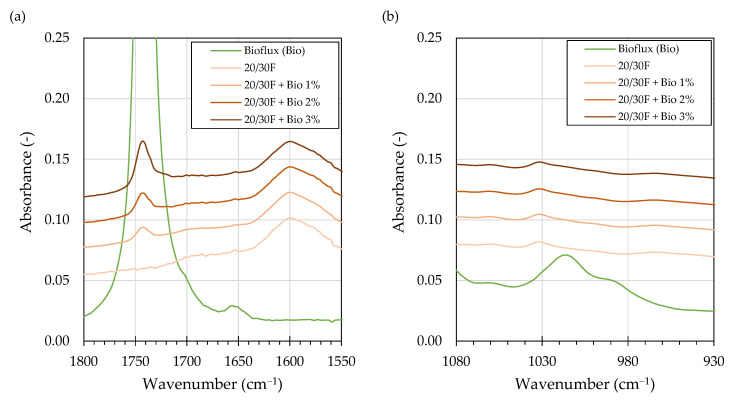
FTIR spectrum overlay of selected foamed (F) asphalt binders and the bio-flux additive showing the carbonyl (**a**) and sulfoxide (**b**) regions showcasing the effects of bio-flux content.

**Figure 4 materials-16-00513-f004:**
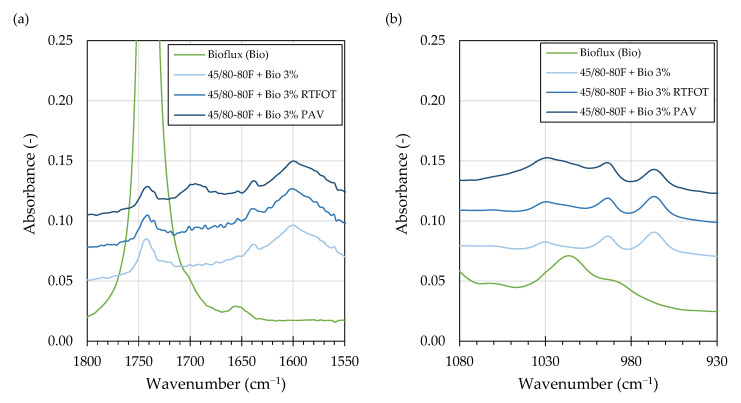
FTIR spectrum overlay of selected foamed (F) asphalt binders and the bio-flux additive showing the carbonyl (**a**) and sulfoxide (**b**) regions showcasing the effects of ageing.

**Figure 5 materials-16-00513-f005:**
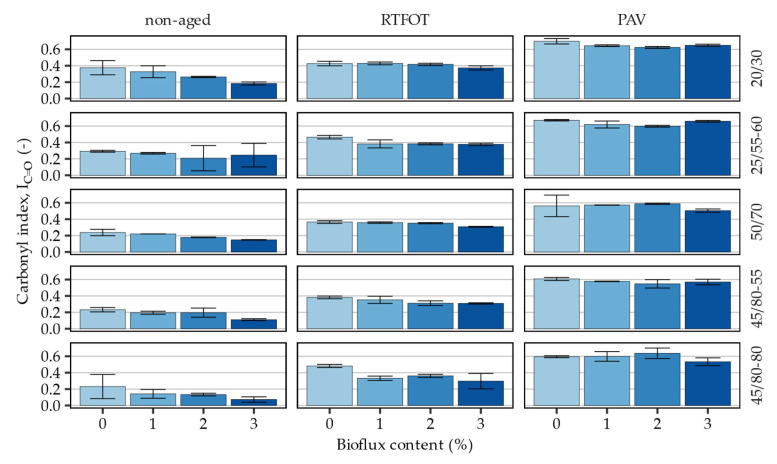
Carbonyl indices calculated based on the FTIR measurements of spectral peaks in the 1700 cm^−1^ wavenumber region (ketone band, 1715–1664 cm^−1^).

**Figure 6 materials-16-00513-f006:**
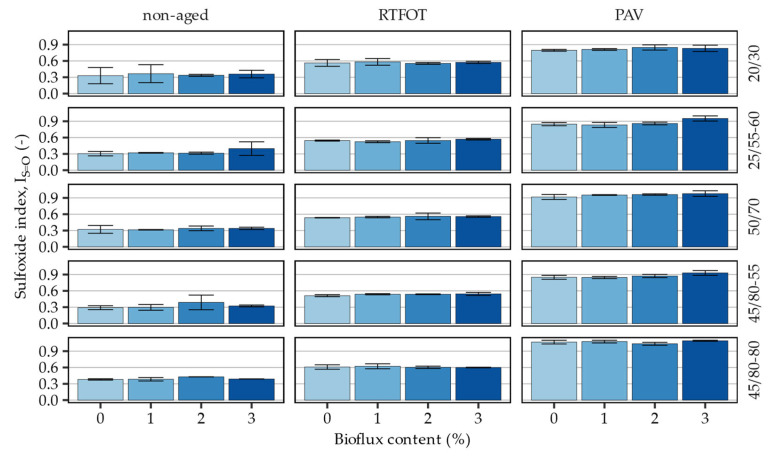
Sulfoxide indices calculated based on the FTIR measurements of 1030 cm^−1^ wavenumber spectral peaks.

**Figure 7 materials-16-00513-f007:**
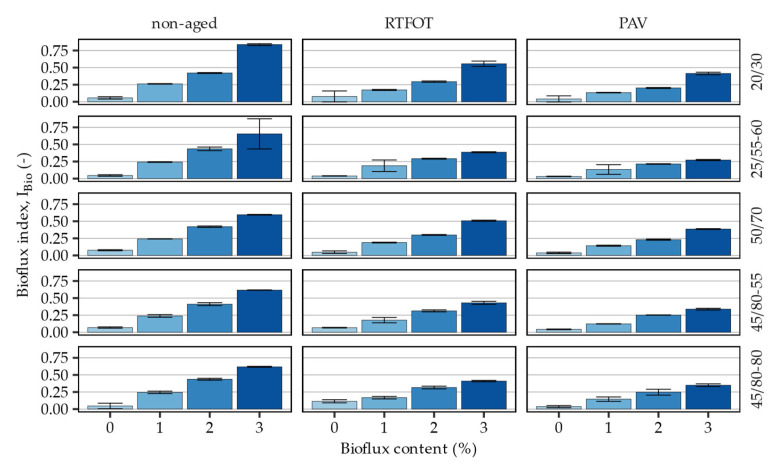
Bio-flux additive related indices calculated based on the FTIR measurements of 1714 cm^−1^ wavenumber spectral peaks.

**Figure 8 materials-16-00513-f008:**
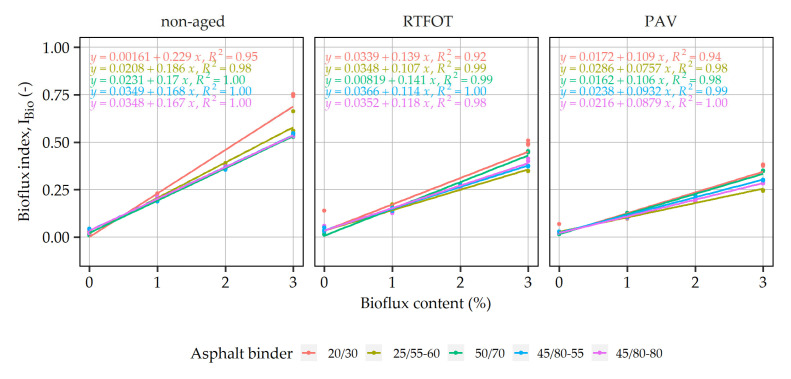
Bio-flux related indices calculated based on the FTIR measurements of 1714 cm^−1^ wavenumber spectral peaks versus added bio-flux additive content.

**Table 1 materials-16-00513-t001:** Properties of the base asphalt binders used in the study.

Property	Unit of Measurement	Base Asphalt Binders					Testing Method
		20/30	50/70	25/55–60	45/80–55	45/80–80	
Penetration at 25 °C	0.1 mm	26	65	40	71	75	EN 1426
Softening point	°C	62.00	48.2	63.4	57.8	95.5	EN 1427
Fraass breaking point	°C	−11	−13	−13	−18	−22	EN 12593
Dynamic viscosity at 135 °C	Pa·s	1.38	0.45	1.70	1.11	2.81	EN 13702-2
Dynamic viscosity at 135 °C after RTFOT	Pa·s	2.31	0.65	2.51	1.51	3.77	EN 13702-2
Elastic recovery after RTFOT	%	-	-	65	83	92	EN 13398

**Table 2 materials-16-00513-t002:** Selected properties of the pure RME for producing the bio-flux additive.

Property	Rapeseed Methyl Esters, RME
Iodine number, g I_2_/100 g	≥100
Viscosity at 25 °C, Pa·	≤0.008
Acid number, mg KOH/g	≤0.50
Flashpoint, °C	≥180

**Table 3 materials-16-00513-t003:** Analysis of the selected reference calculation approaches for assessment of FTIR spectra.

Reference Calculation Approach	Wavenumbers (cm^−1^)	Asphalt Binder	Area, Mean (-)	max−min (-)	Standard Deviation (-)	Coefficient of Variance (%)
(1) Aliphatic group	1525–1350 * [63]	20/30	7.815	0.520	0.066	0.85%
		25/55–60	7.886	0.748	0.107	1.36%
		50/70	7837	0.242	0.048	0.61%
		45/80–55	7.876	0.455	0.084	1.06%
		45/80–80	7.990	0.595	0.212	2.65%
(2) Sum of peaks	2000–600 * [58]	20/30	10.734	0.623	0.180	1.68%
		25/55–60	10.643	1.131	0.218	2.05%
		50/70	10.579	0.693	0.167	1.58%
		45/80–55	10.473	0.840	0.213	2.04%
		45/80–80	10.357	1.224	0.340	3.28%
(3) Sum of peaks	2953–699 * [52]	20/30	50.863	1.065	0.162	0.32%
		25/55–60	51.184	0.779	0.185	0.36%
		50/70	51.022	0.625	0.132	0.26%
		45/80–55	51.207	0.996	0.205	0.40%
		45/80–80	51.635	1.331	0.394	0.76%

* considered peaks: A_(2953, 2923, 2862)_, A_1700_, A_1600_, A_1460_, A_1376_, A_1310_, A_1030_, A_990_, A_966_, A_910_, A_864_, A_814_, A_743_, A_724_, A_699._

**Table 4 materials-16-00513-t004:** Chemical indices calculated for the bituminous binders [52,59,64,65,66,67].

Chemical Index	Chemical Bond, Excitation Mode	Peak Wavenumber (Absorption Band), cm^−1^	Chemical Index Expression:
Ester (bio-flux indicator)	C=O, stretching	1741 (1755–1730)	$I_{Bio}=\frac{A_{1741}}{\sum A_{all}}$
Carbonyl (ketones)	C=O, stretching	1700 (1715–1664)	$I_{C=O}=\frac{A_{1700}}{\sum A_{all}}$
Sulfoxide	S=O, stretching	1030 (1045–986)	$I_{S=O}=\frac{A_{1030}}{\sum A_{all}}$

ΣA_all_ = A_(2953, 2923, 2862)_ + A_1741_ +A_1700_ + A_1600_ + A_1460_ + A_1376_ + A_1310_+ A_1030_ + A_990_ + A_966_ +A_910_ + A_864_ + A_814_ + A_743_ + A_724_ + A_699_

**Table 5 materials-16-00513-t005:** Analysis of variance table for evaluating the effects of bio-flux additive content and laboratory ageing n the measured values of carbonyl index (I_C=O_) in the investigated binders.

Asphalt Binder	20/30		25/55–60		50/70		45/80–55		45/80–80	
Model Parameters: I_C=O_	Effect Estimate	*p*-Value	Effect Estimate	*p*-Value	Effect Estimate	*p*-Value	Effect Estimate	*p*-Value	Effect Estimate	*p*-Value
Intercept:	0.3838	<0.001	0.2839	<0.001	0.2430	<0.001	0.2389	<0.001	0.2170	<0.001
Bio-flux additive:	−0.0644	<0.001	−0.0201	<0.001	−0.0312	<0.001	−0.0378	<0.001	−0.0483	<0.001
Ageing:										
RTFOT	0.0524	<0.001	0.1618	<0.001	0.1297	<0.001	0.1400	<0.001	0.2296	<0.001
PAV	0.2987	<0.001	0.3616	<0.001	0.3381	<0.001	0.3575	<0.001	0.3953	<0.001
Bio-flux										
additive * ageing:										
RTFOT	0.0479	<0.001	−0.0081	<0.001	0.0133	<0.001	0.0111	<0.001	−0.0044	<0.001
PAV	0.0461	<0.001	0.0141	<0.001	0.0154	<0.001	0.0238	<0.001	0.0337	<0.001
Adj. R^2^	0.9826		0.9536		0.9795		0.9869		0.9577	

asterisk (*) denotes interaction.

**Table 6 materials-16-00513-t006:** Analysis of variance table for evaluating the effects of bio-flux additive content and laboratory ageing n the measured values of sulfoxide index (I_S=O_) in the investigated binders.

Asphalt Binder	20/30		25/55–60		50/70		45/80–55		45/80–80	
Model Parameters: I_S=O_	Effect Estimate	*p*-Value	Effect Estimate	*p*-Value	Effect Estimate	*p*-Value	Effect Estimate	*p*-Value	Effect Estimate	*p*-Value
Intercept	0.3395	<0.001	0.2930	<0.001	0.3167	<0.001	0.2978	<0.001	0.3849	<0.001
Bio-flux additive	0.0054	0.527	0.0271	0.001	0.0084	0.051	0.0162	0.026	0.0068	<0.001
Ageing:										
RTFOT	0.2294	<0.001	0.2406	<0.001	0.2219	<0.001	0.2190	<0.001	0.2302	<0.001
PAV	0.4596	<0.001	0.5286	<0.001	0.6085	<0.001	0.5365	<0.001	0.6779	<0.001
Bio-flux										
additive * ageing:										
RTFOT	−0.0049	0.675	−0.0185	0.073	−0.0006	0.913	−0.0058	0.558	−0.0114	0.123
PAV	0.0095	0.412	0.0068	0.514	0.0106	0.078	0.0106	0.303	−0.0036	0.626
Adj. R^2^	0.9721		0.9843		0.9963		0.9846		0.9952	

asterisk (*) denotes interaction.

**Table 7 materials-16-00513-t007:** Analysis of variance table for evaluating the effects of bio-flux additive content and laboratory ageing n the measured values of bio-flux related indices (I_Bio_) in the investigated binders.

Asphalt Binder	20/30		25/55–60		50/70		45/80–55		45/80–80	
Model Parameters: I_Bio_	Effect Estimate	*p*-Value	Effect Estimate	*p*-Value	Effect Estimate	*p*-Value	Effect Estimate	*p*-Value	Effect Estimate	*p*-Value
Intercept	−0.0935	0.059	−0.0339	0.206	0.0060	0.755	0.0123	0.360	−0.0025	0.821
Bio-flux additive	0.0256	<0.001	0.3277	<0.001	0.3151	<0.001	0.3224	<0.001	0.3408	<0.001
Ageing:										
RTFOT	0.0410	0.523	0.0134	0.706	−0.0426	0.125	−0.0251	0.171	−0.0245	0.127
PAV	0.0941	0.153	0.0168	0.653	−0.0450	0.100	−0.0386	0.068	−0.0150	0.346
Bio-flux										
additive * ageing:										
RTFOT	−0.1926	<0.001	−0.1591	<0.001	−0.0791	<0.001	−0.1164	<0.001	−0.1194	<0.001
PAV	−0.3744	<0.001	−0.2241	<0.001	−0.1414	<0.001	−0.1716	<0.001	−0.1914	<0.001
Adj. R^2^	0.9314		0.9649		0.9827		0.9923		0.9946	

asterisk (*) denotes interaction.

## Data Availability

Data available on request from the corresponding author.

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
