# Peer review of "Effects of Laboratory Ageing on the FTIR Measurements of Water-Foamed Bio-Fluxed Asphalt Binders"

_materials, 2023, doi:10.3390/ma16020513_

Round 1

Reviewer 1 Report

FTIR based investigation was performed on bio-based additives. Which is interesting. But the scope of the experiments seems very small.

I would suggest revising the introduction to make it more coherent to the scope of the study (other FTIR studies and discussion of similar functional groups from other construction materials: there have a lot of studies done). Else reading it gives a feel that, authors would focus not only FTIRs but production and compaction temperatures as well as may have done some other chemical analysis as well.

secondly, in figure 3, please Arrange only one type of binder with its unaged, RTFO and then RTFO+PAV only, and repeat for others separately like (a), (b) and so on. 

Author Response

Dear Reviewer,

We would like to voice our gratitude for the time and effort spent revising our paper titled “Effects of laboratory ageing on the FTIR measurements of water-foamed bio-fluxed asphalt binders”. We truly feel that your remarks are a significant contribution to the overall quality of our paper and enabled us to rectify its shortcomings.

After a thorough revision, we present you the corrected version of the manuscript for its assessment. Please find the detailed responses to your comments below.

Best regards,

Authors

Reviewer 1

This study investigated the aging behavior of foamed asphalts containing bio-additives through FTIR spectra, and some specific comments were given below:

1) What do authors mean by “oxidative stress” in Abstract? It is not normally used, and “oxidation” or “thermal oxidation” may be better.

Thank you for the comment. The term was changed to “oxidation” as suggested.

2) The number of keywords should be reduced to no more than five.

Thank you for the comment. The official template for papers submitted to Materials journal states the following: „List three to ten pertinent keywords specific to the article yet reasonably common within the subject discipline.”. Therefore, the seven provided keywords are in line with the journal’s guidelines.

3) Is the so-called “fluxing agent” a widely-used term in the field of paving asphalt materials? To my best knowledge, it seems that no literature has used the term expect for authors. Also please clarify the meaning of “fluxing” in the abstract and introduction.

We appreciate the reviewer for this remark. The term "fluxing agent" was changed in the text to "bio-flux additive" which better describes their nature and behaviour in bitumen. The terminology was unified in the manuscript. Additional explanation was added to the abstract as follows:

"The use of the plant origin additive, in particular oil derivatives, was aimed at softening and better foaming of asphalt binders. This modification is possible due to the good mixability of vegetable oils with asphalt binder, which gives a homogeneous product with reduced stiffness".

4) Specific test parameters of FTIR spectra measurement were missing. Also, how many replicates were performed for each sample?

Thank you for the valuable comment. We have added the requested information in section 2.2.2. FTIR measurements:

“The spectrograms were recorded using 32 scans per sample at 4 cm-1 resolution and three replicates were tested for each evaluated experiment.”

5) Legends were missing in Figure 3(a).

Thank you for the comment, the figures have been corrected.

Reviewer 2 Report

The authors conduct research in the current direction, the research is done at a good experimental level, there are several comments.

1. In the introduction part, there is no mention of the works in which the FTIR spectroscopy method was used for such studies (crude oil, asphaltenes and other petroleum products). Although there are links in the text of the article.

2. It is not clear why the authors used chemical index, where Aall contains a lot of spectral bands, for example, the authors of work (dx.doi.org/10.1021/ef200373v | Energy Fuels 2011, 25, 3552–3567) for the index of carbonyl content is different. What is the need to use bands that essentially duplicate the content of groups, for example bands (3000-2800 cm-1) and bands (1460-1350 cm-1) correspond to the stretching vibrations of C-H and bending vibrations of CH2-CH3 groups.

3. It would be nice if the authors proposed some mechanism for the influence of the additive on the aging process. Why does the aging process decrease when adding Bioflux  agent?

4. In Table 4, replace S=O with C=O for esters.

Author Response

Dear Reviewer,

We would like to voice our gratitude for the time and effort spent revising our paper titled “Effects of laboratory ageing on the FTIR measurements of water-foamed bio-fluxed asphalt binders”. We truly feel that your remarks are a significant contribution to the overall quality of our paper and enabled us to rectify its shortcomings.

After a thorough revision, we present you the corrected version of the manuscript for its assessment. Please find the detailed responses to your comments below.

Best regards,

Authors

Reviewer 2

The authors conduct research in the current direction, the research is done at a good experimental level, there are several comments.

Thank you for the generous comment.

  1. In the introduction part, there is no mention of the works in which the FTIR spectroscopy method was used for such studies (crude oil, asphaltenes and other petroleum products). Although there are links in the text of the article.

Thank you for the accurate and valuable remark. We have supplemented the introduction accordingly (L100-112).

  1. It is not clear why the authors used chemical index, where Aall contains a lot of spectral bands, for example, the authors of work (dx.doi.org/10.1021/ef200373v | Energy Fuels 2011, 25, 3552–3567) for the index of carbonyl content is different. What is the need to use bands that essentially duplicate the content of groups, for example bands (3000-2800 cm-1) and bands (1460-1350 cm-1) correspond to the stretching vibrations of C-H and bending vibrations of CH2-CH3 groups.

Thank you for the remark. In our work we have carefully studied the possibilities for calculating the chemical indices for tracking the changes in the chemical components of investigated asphalt binders. We based our assessment on the the published results of other researchers and analysis provided in section 3.2.1. Other authors have used different approaches in this scope:

  • Mouillet et al. [1] have included in their reference area the 1700, 1600, 1460, 1376, 1030, 864, 814, 743, 724 [cm-1] in calculations of different chemical indices, including carbonyl and sulfoxide indices similar as in our work.
  • Feng et al. [2] as well as Yut and Zofka [3,4] as in our work have used the 2924 – 723 peaks for the calculation of reference areas,
  • Yao et al. [5] have used the 2000 - 600 cm-1 peaks in their analyses,
  • Camargo et al. [6] have used a narrow band of 1525 - 1350 cm-1 for this purpose.

In section 3.2.1 we have analyzed thoroughly three different approaches [4–6]  for assuming the reference areas for the calculation of chemical indices that would return unbiased results regarding the ageing processes and the bio-flux used and we have found that the least bias in the values in the values of the calculated indexes would be obtained when the sum of areas under the 2953 - 699 cm-1 wavenumber peaks are used.

In response to the remark we have also clarified lines 266-267:

“Based on these results, in this study the approach utilizing the 2953-699 cm−1 wavenumber range was selected to obtain the unbiased values  of the calculated chemical indices.”

References cited above:

  1. Mouillet, V.; Lamontagne, J.; Durrieu, F.; Planche, J.-P.; Lapalu, L. Infrared microscopy investigation of oxidation and phase evolution in bitumen modified with polymers. Fuel 2008, 87, 1270–1280, doi:10.1016/j.fuel.2007.06.029.
  2. Feng, Z.G.; Wang, S.J.; Bian, H.J.; Guo, Q.L.; Li, X.J. FTIR and rheology analysis of aging on different ultraviolet absorber modified bitumens. Constr. Build. Mater. 2016, 115, 48–53, doi:10.1016/j.conbuildmat.2016.04.040.
  3. Yut, I.; Zofka, A. Attenuated total reflection (ATR) fourier transform infrared (FT-IR) spectroscopy of oxidized polymer-modified bitumens. Appl. Spectrosc. 2011, 65, 765–770, doi:10.1366/10-06217.
  4. Yut, I.; Zofka, A. Correlation between rheology and chemical composition of aged polymer-modified asphalts. Constr. Build. Mater. 2014, 62, 109–117, doi:10.1016/j.conbuildmat.2014.03.043.
  5. Yao, H.; Dai, Q.; You, Z. Fourier Transform Infrared Spectroscopy characterization of aging-related properties of original and nano-modified asphalt binders. Constr. Build. Mater. 2015, 101, 1078–1087, doi:10.1016/j.conbuildmat.2015.10.085.
  6. Camargo, I.G.D.N.; Hofko, B.; Mirwald, J.; Grothe, H. Effect of thermal and oxidative aging on asphalt binders rheology and chemical composition. Materials (Basel). 2020, 13, 1–21, doi:10.3390/ma13194438.

  1. It would be nice if the authors proposed some mechanism for the influence of the additive on the aging process. Why does the aging process decrease when adding Bioflux  agent?

We appreciate the reviewer for this comment. Additional explanation was added to the paragraph 3.2.2. as follows:

“It must be noted that oxidation is the main factor responsible for the aging of asphalt binders. Molecular oxygen can react with the binder's reactive components, leading to the formation of oxygen compounds, among which ketones, carboxylic acids and sulfoxides predominate. The most intense oxidation of asphalt binder occurs during the mixing, production and laying of the asphalt mixture. Temperature and air access have the greatest impact on the rate of oxidation. For example, a temperature increase of 10°C results in a twofold increase in the oxidation rate, so during operations with asphalt at elevated temperatures, aging occurs to a greater extent. Adding bio-flux additive to the asphalt binder reduces the viscosity of the asphalt binder, and the required process temperature can be reduced, which reduces aging. In addition, bio-flux additive contains unsaturated bonds and is susceptible to the oxypolymerization reaction. This results in the fact that molecular oxygen is primarily involved in the oxypolymerization reaction of the reactive bio-additive, which limits the aging of the asphalt binder.”

  1. In Table 4, replace S=O with C=O for esters.

Thank you for the accurate remark. We have also corrected the numbering of tables.

Reviewer 3 Report

This study investigated the aging behavior of foamed asphalts containing bio-additives through FTIR spectra, and some specific comments were given below:

1) What do authors mean by “oxidative stress” in Abstract? It is not normally used, and “oxidation” or “thermal oxidation” may be better.

2) The number of keywords should be reduced to no more than five.

3) Is the so-called “fluxing agent” a widely-used term in the field of paving asphalt materials? To my best knowledge, it seems that no literature has used the term expect for authors. Also please clarify the meaning of “fluxing” in the abstract and introduction.

4) Specific test parameters of FTIR spectra measurement were missing. Also, how many replicates were performed for each sample?

5) Legends were missing in Figure 3(a).

Author Response

Dear Reviewer,

We would like to voice our gratitude for the time and effort spent revising our paper titled “Effects of laboratory ageing on the FTIR measurements of water-foamed bio-fluxed asphalt binders”. We truly feel that your remarks are a significant contribution to the overall quality of our paper and enabled us to rectify its shortcomings.

After a thorough revision, we present you the corrected version of the manuscript for its assessment. Please find the detailed responses to your comments below.

Best regards,

Authors

Reviewer 3

FTIR based investigation was performed on bio-based additives. Which is interesting. But the scope of the experiments seems very small.

Thank you for the remark.

I would suggest revising the introduction to make it more coherent to the scope of the study (other FTIR studies and discussion of similar functional groups from other construction materials: there have a lot of studies done). Else reading it gives a feel that, authors would focus not only FTIRs but production and compaction temperatures as well as may have done some other chemical analysis as well.

Thank you for the accurate and valuable remark. We have supplemented the introduction accordingly (L100-112).

secondly, in figure 3, please Arrange only one type of binder with its unaged, RTFO and then RTFO+PAV only, and repeat for others separately like (a), (b) and so on. 

Thank you for the remark. The figure has been separated into two, each showing different effects.

Round 2

Reviewer 3 Report

The authors have carefully revised the manuscript and responded well to my comments. The revised manuscript can be accepted. Congratulations!